# Towards a Uniform Welding Quality: A Novel Weaving Welding Control Algorithm Based on Constant Heat Input

**DOI:** 10.3390/ma15113796

**Published:** 2022-05-26

**Authors:** Hong Lu, Zidong Wu, Yongquan Zhang, Yongjing Wang, Shu Liu, He Huang, Meng Liu, Shijie Liu

**Affiliations:** 1School of Mechanical and Electronic Engineering, Wuhan University of Technology, Wuhan 430070, China; landzh@whut.edu.cn (H.L.); wuzidong@whut.edu.cn (Z.W.); huangh0513@163.com (H.H.); liumenghn@whut.edu.cn (M.L.); shijieliu6@126.com (S.L.); 2Department of Mechanical Engineering, School of Engineering, University of Birmingham, Birmingham B15 2TT, UK; y.wang@bham.ac.uk; 3China Construction Science and Industry Corporation Limited, Wuhan 430070, China; liushu@cscec.com

**Keywords:** weaving weld, multi-pass weld, constant heat input, velocity-adaptive trajectory planning algorithm

## Abstract

The weaving welding process is a key method used to improve the welding quality in multi-layer and multi-pass welding processes using robots. However, the heat-input fluctuation in the weaving welding process restricts its development. In this paper, we developed a novel weaving welding control algorithm to maintain a constant weld heat input through velocity adaptive planning. First, the heat consumption during the weaving welding was modeled to describe the influence of weaving parameters on the weld heat input. Then, based on the obtained relationship between the weld heat input and the weaving parameters, a velocity-adaptive trajectory planning strategy was proposed by leveraging the transformation matrix derived from the relationship between the workpiece and the robot co-ordinate systems. The simulation and experimental results show that the proposed strategy can compensate for the weaving parameters to maintain a constant heat input based on heat consumption and improve the quality of the robotic multi-layer and multi-pass welding process.

## 1. Introduction

With the developments in information sensing and modern manufacturing technologies, it has become a major trend to realize automatic, flexible and intelligent welding manufacturing [1,2]. Research and development of welding robots related to algorithm design, control systems and sensing ability is emphasized by engineers as a potential area because it has a significant and positive impact on overall industrial production processes [3,4]. Automated intelligent welding robots rely on control algorithm design, which is usually defined as the establishment of a new control algorithm using mathematical models [5,6].

Weaving welding is a welding operation in which the weld heat source performs regular transverse weaving on the weldment during welding in order to cover a larger surface area [7,8] and is used frequently because of its efficiency, adaptability and stability [9,10]. Chang et al. [11] proposed a seam-tracking algorithm based on characteristic point detection. The algorithm focuses on the robot weaving weld because the high efficiency and environmental adaptability of weaving welding. Wang et al. [12] proposed a virtual reality human–robot collaborative welding system, which also considers the robot weaving weld because of the robustness of weaving welding. Furthermore, the weaving parameters are important influence factors of welding quality. Wu et al. [13] researched dynamic characteristics of weaving parameters of arc-weaving P-GMAW and observed the dynamic behavior of welding droplet transfer. Xu et al. [14] established a 3D model to simulate dynamic characteristics of a motel pool in the process of swing-arc GMA welding. They researched the mechanism of weld formation in weaving welding and proposed a weaving frequency range to obtain a satisfactory weld. Typically, welding parameters are usually investigated in the literature. Chen et al. [15] focused on the effect of weaving frequency and amplitude on the temperature field in the weaving welding process. Although previous methods expound on some major parameters, none supports the influence of the inclination of the welding torch. The welding requirements for uniform welding quality under complex conditions are difficult to achieve.

Control algorithms for weaving welding are developed to avoid deterioration of welding quality. Efficient and intelligent control algorithms can ensure reliable and high-quality automatic weaving welding. Kang et al. [4] proposed a control algorithm of circular trajectory weaving welding based on the space transformation principle. The algorithm calculates the weaving path and weaving parameters according to the initial path and spatial transformation relationship. Gao et al. [16] proposed a novel welding planning algorithm to solve the continuous welding collision-free problem. The algorithm effectively uses welding angular redundancy to implement welding torch swinging and obstacle avoidance motion planning. Liu et al. [17] proposed a robotic weaving welding planning algorithm for multi-layer sphere-pipe joints and established a description model of the welding torch inclination. However, none of these algorithms considers the influence of inclination variation of the welding torch in detail.

Weaving welding technology is usually applied to weld steel structures in architectural applications [18], and it is difficult to automate [19] due to the challenges associated with parameter selection and process control [20]. Research on robotic autonomous welding shows that guaranteeing weld quality in the groove edge area is more difficult than in the central groove area [21,22]. Variation in the inclination of the welding torch causes this issue, making it difficult to ensure weld quality [23]. However, the problem is usually overlooked by current robot welding planning methods. According to the theory presented by Parvez et al. [24], the heat flux generated by conduction and convection is sensitive to and correlated with the torch inclination. Variation in the inclination of the welding torch could lead to poor and uneven welding quality. Furthermore, the finite element (FE) method is usually used to analyze the thermal effect of the welding process [25]. Taraphdar et al. [26] studied the residual stress distribution of different FE models and investigated the accuracy of residual stress distribution prediction. Approaches considering multiple models are worth applying to study welding heat input. Pandey et al. [27] studied the effect of welding direction to minimize distortion in fillet joints and also applied the FE method. Therefore, the FE method is applicable for mechanism and process studies of weaving welding. However, control algorithms for weaving parameters are rarely reported in the literature, especially algorithms considering the influence of heat-input fluctuation [28] produced by welding inclination. Maintaining constant welding heat input in weaving welding remains a challenge [29].

In this paper, we developed a novel weaving welding control algorithm to maintain constant welding heat input through velocity adaptive planning. A heat consumption model of weaving welding was established to describe the influence of weaving parameters on the welding heat input. Then, based on the obtained relationship between the welding heat input and the weaving parameters, a velocity-adaptive trajectory planning strategy was proposed by leveraging the transformation matrix derived from the relationship between the workpiece and the robot co-ordinate systems. In addition, the simulation and experimental results show that the proposed strategy can compensate for the weaving parameters to improve the quality of the robotic multi-layer and multi-pass welding process. The proposed strategy can maintain the uniformity of the welds in the groove edge area and prevent the occurrence of weld defects in the welding of steel structures.

## 2. Heat Consumption during Weaving Welding

Welding heat consumption is related to the weaving parameters, such as velocity, amplitude, period length, etc., resulting in the fluctuation of heat input during the multi-layer and multi-pass welding process. In this section, the theoretical model for heat consumption considering the weaving parameters is derived.

### 2.1. Weaving Welding Parameters

The welding parameters are the key factors to ensure welding flexibility and quality and are determined by the robot [30,31]. According to the forming behavior of the forming of welding pool, the weaving welding parameters using a robot, as shown in Figure 1, are defined as follows.
Welding velocity is defined as the moving velocity of the end point of the welding wire, which is controlled by the inclination of the welding torch;Weaving amplitude is defined as the maximum distance from the weaving position to the center line of the weaving path;Weaving period length is defined as the distance of a weaving period along the welding direction per unit time;Dwell time is defined as the time of movement suspension when the end point of the welding wire is at the weaving amplitude point. The dwell time is not always equal on all sides, depending on the welding process.Welding torch inclination is defined as the angle between the axis of welding torch and the vertical direction.

### 2.2. Heat Consumption Model

In the existing literature, the velocity of the end point of the welding wire is assumed to be constant during the weaving welding process. However, the welding torch can rotate, and the inclination, $\varphi_{1}$, increases when the torch moves close to edge of the groove. As a result, the distance between the end point of the welding wire and the welded surface varies, which leads to the degradation of the heat input caused by the variation in the direction of gas shear stress, arc force and the heat flux of the pool surface [24,32]. Additionally, increased inclination of the welding torch causes the profile of the weld pool to become a larger ellipse, as shown in Figure 2. When the distance is large enough, the welding current decreases dramatically, causing welding failure. To maintain a constant heat input, the relationship between the inclination angle of the welding torch and heat consumption during the weaving welding process is explored.

Δ*E* is defined as the heat input of the molten pool at a given time during the vertical motion process of the welding torch. The decreased heat input of the molten pool is a function of the rotation angle, $\varphi_{1}$. Assuming the function is $f\left(\varphi_{1}\right)$, the produced heat input of the molten pool at a given time during the tilting motion process of the welding torch, $\Delta E_{d}$, can be expressed as:(1)$$\Delta E_{d}=\Delta E-f\left(\varphi_{1}\right)$$

The heat input per unit area of the welded surface at a given time, $\Delta E_{a}$, can be expressed as:(2)$$\Delta E_{a}=\frac{E_{ut}}{S_{p}}$$
where $E_{ut}$ is the total heat input of the molten pool at a given time, and $S_{p}$ is the area of the arc section on the welded surface.

As shown in Figure 2, the molten pool on the welded surface can be described as an ellipse. During the vertical motion process of the welding torch, the major and minor axes are $l_{a}$ and $r_{a}$, respectively. Regarding the rotated welding torch, if the decrease in the heat input of molten pool is neglected, the minor axis of the sectional ellipse is equal to 2$r_{a}$, and the major axis of the sectional ellipse becomes $l_{m}$. Finally, the major and minor axes of the ellipse would be $l_{p}$ and 2$r_{p}$, respectively, considering the decrease in heat input.

According to the geometrical relationship shown in Figure 2, the major axis of the sectional ellipse, $l_{m}$, can be expressed as:(3)$$l_{m}=\frac{2r_{a}}{\cos\varphi_{1}}$$

The ratio of $S_{p}$ and $S_{a}$, which is defined as $e\left(\varphi_{1}\right)$, can be derived as:(4)$$\begin{array}{l} \frac{S_{p}}{S_{a}}\propto e\left(\varphi_{1}\right)\\ \Rightarrow \frac{\pi g^{2}r_{a}\left(\frac{l_{m}}{2}\right)}{\pi r_{a}\left(\frac{l_{m}}{2}\right)}\propto e\left(\varphi_{1}\right)\\ \Rightarrow g^{2}\propto e\left(\varphi_{1}\right) \end{array}$$

According to Equation (4), *g* can be expressed as:(5)$$g=\lambda \sqrt{e\left(\varphi_{1}\right)}$$
where *λ* is the proportionality factor.

Therefore, the heat input per unit area of the welded surface at a given time during the vertical motion process and the tilted motion process of the welding torch, $\Delta E_{av}$, and $\Delta E_{at}$, respectively, can be expressed as Equations (6) and (7).
(6)$$\Delta E_{av}=\frac{\Delta E}{S_{pc}}=\frac{\Delta E}{\pi \left(\frac{l_{a}}{2}\right)r_{a}}$$
(7)$$\Delta E_{at}=\frac{\Delta E_{d}}{S_{po}}=\frac{\Delta E-f\left(\varphi_{1}\right)}{\pi \left(\frac{l_{p}}{2}\right)r_{p}}$$
where $S_{pc}$ is the area of the molten pool section on the welded surface during the vertical motion process of the welding torch, and $S_{pt}$ is the sectional area of the molten pool on the welded surface during the tilted motion process of the welding torch.

As the total time of heat input per unit area is equivalent to the time the molten pool goes through the unit point on the welded surface, $t_{total}$ can be expressed as:(8)$$t_{total}=\frac{l}{v}$$
where *v* is the velocity of the welding torch.

The total input heat per unit area, $E_{t}$, can be expressed as:(9)$$E_{t}=\Delta E_{a}t_{total}$$

Based on Equation (9), the total input heat per unit area when the welding torch is vertical and tilting, $\Delta E_{tv}$ and $\Delta E_{tt}$, respectively, can be expressed as:(10)$$E_{tv}=\Delta E_{av}t_{v}=\frac{\Delta E}{\frac{1}{2}\pi r_{a}v}$$
(11)$$E_{tt}=\Delta E_{at}t_{t}=\frac{\Delta E-f\left(\varphi_{1}\right)}{\left(\frac{1}{2}\right)\pi r_{p}v}$$

Then, based on the Equations (3), (5), (10) and (11), the relationship of $E_{tt}$ and $E_{tv}$ can be derived as:(12)$$E_{tt}=\frac{\sqrt{e\left(\varphi_{1}\right)}}{\lambda }E_{tv}$$

Considering that heat input decreases when the tilting angle increases, the heat input of the weld arc is assumed to be proportional to the length of the weld arc. Therefore, $e\left(\varphi_{1}\right)$ can be expressed as $\cos\varphi_{1}$, and $E_{tt}$ can be expressed as:(13)$$E_{tt}=\sqrt{\cos\varphi_{1}}E_{tv}$$
where *λ* is equal 1 when the welding torch is vertical.

According to the above derivations, the correlation of the variation in the heat input with the inclination angle is shown in Figure 3. When the welding torch is vertical, $\varphi_{1}$ is equal to zero and $E_{tt}$ is equal to $E_{tv}$. When the tilting angle of the welding torch is larger than the available welding inclination, $E_{tt}$ is low, and the welding process is interrupted.

Based on Equation (13), the quality of consumed heat can be represented as the difference between $E_{tv}$ and $E_{tt}$. The heat consumption during the welding process, $E_{c}$, can be expressed as:(14)$$E_{c}=\left(1-\sqrt{\cos\varphi_{1}}\right)E_{tv}$$

## 3. Velocity-Adaptive Trajectory Planning Strategy for Multi-Pass Welding

According to the established heat consumption model, a novel trajectory planning method to maintain constant heat input based on the adaptive adjustment of the welding velocity is proposed in this section.

### 3.1. Robotic Welding Systems

There are four co-ordinate systems in the robotic welding system. Generally, the relationship between each co-ordinate system determines the actual position of the welding torch motion in robotic weaving welding planning. The co-ordinate systems are defined as follows:$\left\{S_{g}:O_{g}-X_{g}Y_{g}Z_{g}\right\}$ is the reference co-ordinate system associated with the welding groove, and it is used to represent the dimension and location of the groove, also called groove co-ordinate system.$\left\{S_{b}:O_{b}-X_{b}Y_{b}Z_{b}\right\}$ is the basic co-ordinate system of the robotic welding system, which refers to the center of the base of the robot. It corresponds to the origin of the robotic welding program.$\left\{S_{e}:O_{e}-X_{e}Y_{e}Z_{e}\right\}$ is the robotic co-ordinate system, which represents the trajectory of the end effector of the robot.$\left\{S_{t}:O_{t}-X_{t}Y_{t}Z_{t}\right\}$ is the tool center point of the robotic welding system, which is fixed on the end point of the welding wire. It represents the location and orientation of the welding torch and is also called welding torch co-ordinate system.

The profile of the groove is described in the groove co-ordinate system. The relationship between the robotic end co-ordinate system and the basic co-ordinate system is based on the structure of the robot. Additionally, the relationship between the welding torch co-ordinate system and the robotic end co-ordinate system is based on the size of the welding torch and the position of clamping. For an arbitrary robot and welding torch, the trajectory of the end of the welding torch in the basic co-ordinate system is determined by the trajectory of the welding torch co-ordinate system, which can be obtained by transformation matrix based on robotic kinematics and TCP selection. The welding trajectory can be represented in the basic co-ordinate system. Therefore, to simplify the transformation relationship of the co-ordinate system, the calculation of the relationship between the groove co-ordinate and basic co-ordinate systems, as well as the dimensions of the groove, is represented in the basic co-ordinate system. Finally, the trajectory of the welding torch and the dimensions of the groove are both accounted for in the basic co-ordinate system, and the relationship between them can be obtained.

The experimental system mainly consists of three parts: a welding robot, a vision sensor and an industrial personal computer (IPC), as presented in Figure 4. The vision sensor is rigidly fixed on the robotic manipulator, forming an eye-in-hand system to detect the groove. Calculations of the co-ordinate system transformation and welding speed planning are performed by the industrial computer. As a motion actuator, the welding robot receives the transformed co-ordinates of the welding seam from the industrial computer.

The standard groove is combined with three planes: the left side plane, the right side plane and the basic button plane [33] (Figure 5a). Eight points of the standard groove are defined as follows: point $P_{f1}$ to $P_{f4}$ in front and point $P_{b1}$ to $P_{b4}$ in the back. Initially, the basic co-ordinate system is defined as shown in Figure 5b. The weaving motion is described in the groove co-ordinate system, in which point $P_{f2}$ is selected as the origin. The weaving path is solved in the groove co-ordinate system according to the welding parameters is then mapped to the real robot motion co-ordinate system and the basic co-ordinate system by the transformation.

As shown in Figure 5b, the transformation matrix from the basic co-ordinate system to the groove co-ordinate system can be expressed as Equation (15) [4].
(15)$$T_{bg}=\left(\begin{array}{cc} R_{bg} & Tr_{bg}\\ 0 & 1 \end{array}\right)$$
where $R_{bg}$ is the rotation matrix, and $Tr_{bg}$ is the translation matrix.

Point $P_{f2}$ is selected as the original point of the groove co-ordinate system. $O_{g}$, so the co-ordinate of the original point is equal to the co-ordinate of point $P_{f2}$ in the basic co-ordinate system. $R_{bg}$ and $Tr_{bg}$ can be expressed as Equations (16) and (17), respectively.
(16)$$R_{bg}=\left(\begin{array}{ccc} \vec{x_{g}} & \vec{y_{g}} & \vec{z_{g}} \end{array}\right)$$
(17)$$Tr_{bg}={\left(\begin{array}{ccc} x_{P_{f2}} & y_{P_{f2}} & z_{P_{f2}} \end{array}\right)}^{T}$$
where $\vec{x_{g}}$, $\vec{y_{g}}$ and $\vec{z_{g}}$ are the direction unit vectors of the $X_{b}$ axis, $Y_{b}$ axis and $Y_{b}$ axis, respectively; and $x_{p_{f2}}$, $y_{P_{f2}}$ and $z_{P_{f2}}$ are the co-ordinates of point $P_{f2}$ in the basic co-ordinate system.

$P_{f2}\left(x_{f2},y_{f2},z_{f2}\right)$, $P_{f3}\left(x_{f3},y_{f3},z_{f3}\right)$ and $P_{b2}\left(x_{b2},y_{b2},z_{b2}\right)$ are defined as the co-ordinates of points $P_{f2}$, $P_{f3}$ and $P_{b2}$, respectively, in the basic co-ordinate system. For the groove co-ordinate system, the direction of $X_{g}$ is the same as vector $\vec{P_{f3}O_{g}}$, and the direction of $Y_{g}$ is the same as vector $\vec{O_{g}P_{b2}}$. $\vec{P_{f3}O_{g}}$ and $\vec{O_{g}P_{b2}}$ can be expressed as Equations (18) and (19), respectively.
(18)$$\vec{P_{f3}O_{g}}=\vec{P_{f3}P_{f2}}={\left(\begin{array}{ccc} x_{f3}-x_{f2} & y_{f3}-y_{f2} & z_{f3}-z_{f2} \end{array}\right)}^{T}$$
(19)$$\vec{O_{g}P_{b2}}=\vec{P_{f2}P_{b2}}={\left(\begin{array}{ccc} x_{f2}-x_{b2} & y_{f2}-y_{b2} & z_{f2}-z_{b2} \end{array}\right)}^{T}$$

Therefore, the direction unit vectors $\vec{x_{g}}$ and $\vec{y_{g}}$ can be expressed as Equations (20) and (21), respectively.
(20)$$\vec{x_{g}}=\frac{\vec{P_{f3}O_{g}}}{\left|\vec{P_{f3}O_{g}}\right|}=\left(\begin{array}{c} x_{x_{g}}\\ y_{x_{g}}\\ z_{x_{g}} \end{array}\right)$$
(21)$$\vec{y_{g}}=\frac{\vec{O_{g}P_{b2}}}{\left|\vec{O_{g}P_{b2}}\right|}=\left(\begin{array}{c} x_{y_{g}}\\ y_{y_{g}}\\ z_{y_{g}} \end{array}\right)$$
where $\vec{z_{g}}$ is the normal vector of plane $X_{g}O_{g}Y_{g}$ and the outer product of $\vec{x_{g}}$ and $\vec{y_{g}}$. $\vec{z_{g}}$ can be expressed as:(22)$$\vec{z_{g}}=\vec{x_{g}}\times \vec{y_{g}}=\left(\begin{array}{c} y_{x_{g}}z_{y_{g}}-y_{y_{g}}z_{x_{g}}\\ z_{x_{g}}x_{y_{g}}-z_{y_{g}}x_{x_{g}}\\ x_{x_{g}}y_{y_{g}}-x_{y_{g}}y_{x_{g}} \end{array}\right)=\left(\begin{array}{c} x_{z_{g}}\\ y_{z_{g}}\\ z_{z_{g}} \end{array}\right)$$

In addition, $\vec{x_{g}}$ and $\vec{y_{g}}$ are orthogonal, and $\left|\vec{z_{g}}\right|$ can be expressed as Equation (23).
(23)$$\left|\vec{z_{g}}\right|=\left|\vec{x_{g}}\right|\left|\vec{y_{g}}\right|sin\theta$$
where *θ* is the included angle of vector $\vec{x_{g}}$ and $\vec{y_{g}}$.

According to Equations (15)–(17) and (20)–(23), the transformation matrix, $T_{bg}$, from the basic co-ordinate system to the groove co-ordinate system can be obtained.
(24)$$T_{bg}=\left(\begin{array}{cccc} x_{x_{g}} & x_{y_{g}} & y_{x_{g}}z_{y_{g}}-y_{y_{g}}z_{x_{g}} & x_{P_{f2}}\\ y_{x_{g}} & y_{y_{g}} & z_{x_{g}}x_{y_{g}}-z_{y_{g}}x_{x_{g}} & y_{P_{f2}}\\ z_{x_{g}} & z_{y_{g}} & x_{x_{g}}y_{y_{g}}-x_{y_{g}}y_{x_{g}} & z_{P_{f2}}\\ 0 & 0 & 0 & 1 \end{array}\right)$$

### 3.2. Velocity-Adaptive Trajectory Planning Algorithm

Considering the heat consumption in welding torch tilting weaving, the aim of velocity planning is to maintain constant heat input, as defined in Equation (25), when the torch inclination increases, which means invariable total heat input in the area of the weld pool of unit length.
(25)$$\Delta E_{tv}=\Delta E_{tt}$$

The weaving welding process simplified to a selected 2D section, which is parallel to plane $X_{g}O_{g}Z_{g}$, is shown in Figure 6. $P_{1}\left(x_{P_{1}},y_{P_{1}},z_{P_{1}}\right)$ and $P_{2}\left(x_{P_{2}},y_{P_{2}},z_{P_{2}}\right)$ are the intersection points of left side plane and the selected section in the groove co-ordinate system.

The selected section can show the motion of the welding torch in the $X_{g}$ and $Z_{g}$ direction, as shown in Figure 6a. When the welding torch moves to the location near the left side plane, interference occurs between the torch and the left side plane if the welding torch is vertical. The boundary condition to avoid interference can be expressed as:(26)$$r_{t}-\Delta l_{l}<\frac{\left(x_{P2}-x_{P1}\right)\left(h_{w}+l_{w}\right)}{z_{P1}-z_{P2}}$$
where $r_{t}$ is the radius of the welding nozzle, $\Delta l_{l}$ is distance between the collision point and $P_{2}$ in the $X_{g}$ direction, $h_{w}$ is the distance from the wire terminal to the basic button plane in the $Z_{g}$ direction and $l_{w}$ is the outstretched length of welding wire.

Incline adjustment of the welding torch in multi-layer welding application is shown in Figure 6b. To realize the collision-free welding path by tilting the welding torch, especially in the area near the left side plane, the boundary condition can be expressed as:(27)$$r_{t}\cos\varphi_{1}-\Delta l_{w}\sin\varphi_{1}<\Delta l_{l}+\Delta l_{g}$$
where $\Delta l_{g}$ is the distance from $P_{2}$ to point *N* along the $X_{g}$ direction, and *N* is the nearest point from *F* to the left side plane along the moving direction of the welding torch.

According to the geometric relationship, $\Delta l_{g}$ can be expressed as:(28)$$\Delta l_{g}=\frac{\left(x_{P2}-x_{P1}\right)h_{F}}{z_{P1}-z_{P2}}$$
where $h_{F}$ is the distance from *F* to the basic button plane in the $Z_{g}$ direction, $X_{P_{1}}$, $Z_{P_{1}}$ are the co-ordinates of point $P_{1}$ and $X_{P_{2}}$, $Z_{P_{2}}$ are co-ordinates of point $P_{2}$.

Based on the geometric relationship, $h_{F}$ can be expressed as:(29)$$h_{F}=h_{w}+l_{w}\cos\varphi_{1}+r_{t}\sin\varphi_{1}$$
where $h_{w}$ is the distance from the end point of the welding wire to the basic button plane in the $Z_{g}$ direction, $l_{w}$ is the outstretched length of the welding wire and $r_{t}$ is the radius of the welding torch nozzle.

According to the heat consumption model proposed in Section 2, when the welding line velocity is constant, the relationship between $\Delta E_{tv}$ and $\Delta E_{tt}$ can be determined. If the heat input is constant, the relationship of $v_{1}$ and $v_{2}$ can be represented as:(30)$$v_{2}=\frac{\sqrt{e\left(\varphi_{1}\right)}}{\lambda }v_{1}$$

Based on Equation (14), $v_{2}$ can be expressed as:(31)$$v_{2}=\sqrt{\cos\varphi_{1}}v_{1}$$

To guarantee the safe motion of the welding torch, a safe distance, $d_{s}$, an extra distance between welding torch and face of groove, is set. According to the related parameters, the $v_{2}$ can be calculated. The position of the end point of the welding wire, $x_{w}$, is equal to $\Delta l_{l}$. Equations (27) and (31) can be rewritten as:(32)$$\left\{\begin{array}{c} r_{t}\cos\varphi_{1}\left(t\right)-l_{w}\sin\varphi_{1}\left(t\right)=x_{w}\left(t\right)+d_{s}\\ v_{2}\left(t\right)=\frac{\sqrt{e\left(\varphi_{1}\left(t\right)\right)}}{\lambda }v_{1}\left(t\right) \end{array}\right.$$

Then, $v_{2}$ can be obtained by the function $f\left(x_{w}\right)$.
(33)$$\left\{\begin{array}{c} v_{2}=\frac{dx_{w}}{dt}=f\left(x_{w}\right)\\ \frac{dx_{w}}{f\left(x_{w}\right)}=dt\\ \int \frac{1}{f\left(x_{w}\right)}dx_{w}=t+C\\ x_{w}=F\left(t\right)\\ v_{2}=\frac{dx_{w}}{dt}=\frac{dF\left(t\right)}{dt} \end{array}\right.$$

In summary, the whole welding planning process using the proposed algorithm can be represented as a flow chart, as shown in Figure 7. Initially, the point cloud of the groove and the weld path are input data. The groove co-ordinate system and transformation relationship can be obtained by the input point cloud, and the transformed path is calculated by transformation relationship. Then, an inclination angle of the welding torch is selected to satisfy the collision-free boundary condition. With the satisfied angle, the welding velocity is calculated by the obtained equations to control the robot.

## 4. Discussion

To verify the correctness of the proposed model and method, a groove weaving welding task was performed. Considering the functionality of the model, we chose a narrow and deep groove.

We used an actual steel test plate as the experimental object, which was equipped with a standard groove. Then, we chose a point cloud obtained by a line structure light sensor to describe the groove because of its accuracy and richness of information [34]. The acquired point cloud is equally spaced; namely, the distance between the *X* direction and *Y* direction is fixed, including depth information of each measured point on the groove. According to the point cloud, the profile of groove can be obtained.

The line structure light sensor moves in a uniform, linear motion, and the camera on the sensor takes pictures with a fixed frequency. The resolutions of the sensors in the *X*, *Y* and *Z* directions are 0.2 mm, 0.5 mm and 0.1 mm, respectively.

The measured points of the groove are reconstructed, as shown in Figure 8. The numbers of sample section and sample point for each section are both 1200. The gray surface is the reconstructed test steel plate and platform under a light source. According to the point cloud, the shape, location and dimension of the groove are obtained to plan the weaving welding path.

However, because of machining errors, the dimensions of the groove are not absolutely standard. Therefore, the dimensions of each section of the groove are not equal, and the face of the groove is not uniform. To overcome the errors, all collected data of each section were adopted to plan the weaving welding path, as shown in Figure 9.

A section of the selected groove is shown in Figure 10. The characteristic points of the groove on a test steel plate are defined, and the groove co-ordinate system is established.

According to the method proposed in Section 3, the process of inclination variation can be obtained, and the parameters of the experimental groove are substituted into Equation (26) to obtain the boundary condition. The relevant parameters of the weaving welding process are calculated as follows. The length of the wire, $l_{w}$, is 15 mm. The radius of welding torch nozzle, $r_{t}$, is 8 mm. $\Delta l_{l}$ at the horizon from the first collision point (approximately equal to $r_{t}$) is equal to 0.5 mm.

As a result, the variation of $\varphi_{1}$ can be obtained, which satisfies the requirement of a safe distance of the welding torch. The set of $\varphi_{1}$ is shown in Figure 11.

According to Equation (32), the calculated $\varphi_{1}$ can ensure the relationship between $v_{2}$ and $v_{1}$. The velocity of the welding torch near the groove area before planning is uniform, as shown in Figure 12a. The relationship between $v_{2}$ and $\varphi_{1}$ is shown in Figure 12b, and the relationship between $\varphi_{1}$ and $\Delta l_{l}$ is obtained. Finally, the relationship between $v_{2}$ and $\Delta l_{l}$ can be calculated and is shown in Figure 12c.

Because the co-ordinates of the torch motion path are determined the groove co-ordinate system, the transformation matrix, $T_{bg}$, can be utilized to transform the co-ordinates of torch motion path into the robotic basic co-ordinate system.

The result of simulation shows the planning process of path and torch inclination. Additionally, the uniform heat input of the whole process can be verified, and the total input heat of the welded arc in a given period of time, $E_{ut}$, can be expressed as:(34)$$E_{ut}=\int_{t_{0}}^{t_{1}}E_{dt}\left(t\right)dt$$
where $t_{0}$ is the start time, $t_{1}$ is the end time and $E_{dt}\left(t\right)$ is the total input heat per unit. time.

According to Equations (2), (3) and (5), $E_{ut}$ can be represented as:(35)$$E_{ut}=\int_{t_{0}}^{t_{1}}E_{dt_{0}}\cos\left[\varphi_{1}\left(t\right)\right]dt$$
where $E_{dt_{0}}$ is the total input heat at $t_{0}$.

Therefore, $E_{ut}$ can be discretized as:(36)$$E_{ut}=\sum_{i=1}^{n}\int_{t_{i}-1}^{t_{i}+dt_{\varphi_{1}}}E_{dt_{0}}\cos\left[\varphi_{1}\left(t\right)\right]dt$$
where n can be ensured by experimentation, $dt_{\varphi_{1}}$ is the time of the welding torch from $t_{0}$ to $t_{1}$.

Because $dt_{\varphi_{1}}$ is a short period of time, $\varphi_{1}$ can be assumed to remain constant in $dt_{\varphi_{1}}$. $\varphi_{1}$ can be represented as:(37)$$\varphi_{1}\left(t\right)=\varphi_{1}\left(t_{0}\right)$$

According to Equations (36) and (37), the total heat input is calculated, and the uniform heat input near the edge of the groove is obtained.

## 5. Experiments

In order to verify the effectivity of the proposed method of weaving welding, a welding experiment using the planned trajectory was performed. An experiment was conducted using a robotic welding system, as shown in Figure 13. The system consists of an integrated robot constructed by six Kollmorgan RGMs, a robotic controller, a PC, a line structure light visual sensor, a welding machine and a welding platform. The welding wire used in the experiment is MG with a diameter of 1.2 mm.

The control algorithm was validated with a chosen welding sample and weaving welding parameters. The welding sample is a steel test plate with a V-groove, and the weaving welding parameters are planned with the proposed algorithm. Then, the co-ordinates of the path are transformed into the basic co-ordinate system. During the welding process, the welding torch weaves regularly along the planned trajectory, and the velocity is varied according to the variation in inclination. The welding result is shown in Figure 14, and the weaving welding parameters are shown in Table 1.

The test plate is welded with five layers of welds. The first layer has one pass of welding, and the other layers have two passes. As a result, the welds are uniformly covered on the groove with high quality by stable weaving, and there is no incomplete fusion, surface porosity or other defects in the welding.

During the weaving welding process, the welding torch inclines, and the heat input decreases near the edge of the groove. Based on the control algorithm, the welding velocity is planned to maintain a constant welding heat input. An exemplary weld and partial features are shown in Figure 15, as well as the decreased speed and uniform welding quality. Partial welding results show that the welds near the edge of the groove are full and uniform, with no defects detected by ultrasonic flaw-detection equipment. In addition, the brown layers on the welds are oxide shells, which can be removed and do not influence the welding quality. The overall results show that weaving welding using the proposed algorithm can effectively ensure uniform welding quality and can be applied in industrial manufacturing.

## 6. Conclusions

In this paper, a new weaving welding control algorithm is proposed that employs velocity adaptive planning to maintain a constant heat input. Heat consumption during the weaving welding is modeled to describe the influence of weaving parameters on the weld heat input. A velocity-adaptive trajectory planning strategy based on the obtained relationship between the weld heat input and the weaving parameters is proposed by using a transformation matrix to control the heat input of the weld pool per unit area. The conclusions from this research are as follows:A heat consumption model is established based on the variation in heat input when the welding torch inclination increases. The established heat consumption model reflects the relationship between welding heat input and weaving parameters; as the welding torch tilts, the inclination angle increases, and the heat input decreases. The detailed relationship and inclination are clarified. The weaving welding parameters are also defined.A robotic welding system is established, and the transformed co-ordinate systems are defined with the transformed relationship. The proposed velocity-adaptive trajectory planning algorithm and strategy are based on the obtained relationship between the weld heat input and the weaving parameters. The algorithm controls the welding velocity according to the torch inclination angle and guarantees a constant welding heat input.The algorithm was verified in an actual welding experiment with five layers of welds. Welding results and partial features are presented. The results show that weaving welding using the proposed velocity-adaptive trajectory planning algorithm can effectively solve the problem of uneven welding quality and can be used in complex weaving welding tasks. The welds in the groove edge area of the groove are uniform and are detected without defects by ultrasonic flaw detection equipment.

## Figures and Tables

**Figure 1 materials-15-03796-f001:**
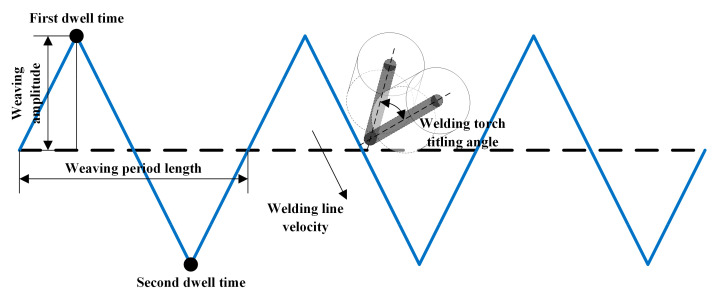
Weaving parameters of weaving welding.

**Figure 2 materials-15-03796-f002:**
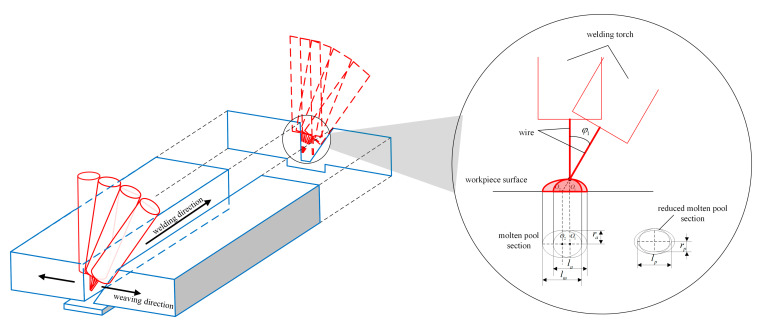
Welding heat-input variation caused by inclination variation.

**Figure 3 materials-15-03796-f003:**
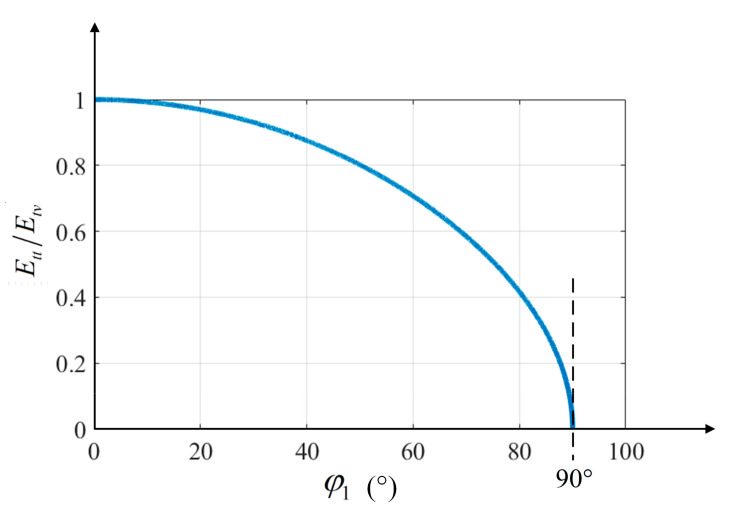
The radio of $E_{tt}$ and $E_{tv}$ with $\varphi_{1}$.

**Figure 4 materials-15-03796-f004:**
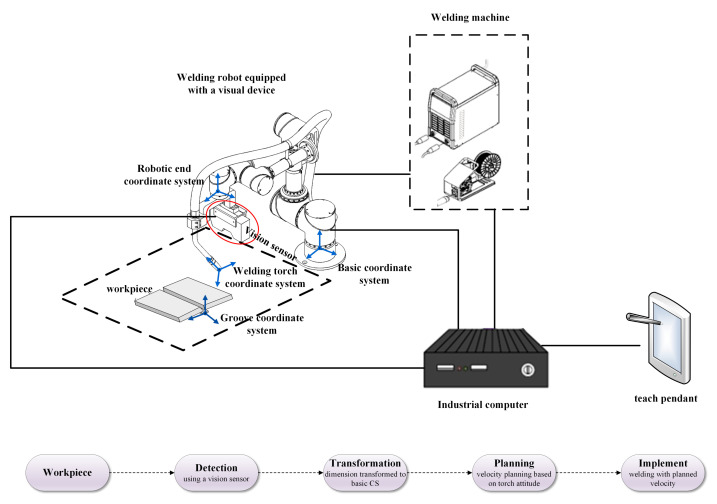
Diagram of the robotic and visual welding system. The co-ordinate systems are represented in blue.

**Figure 5 materials-15-03796-f005:**
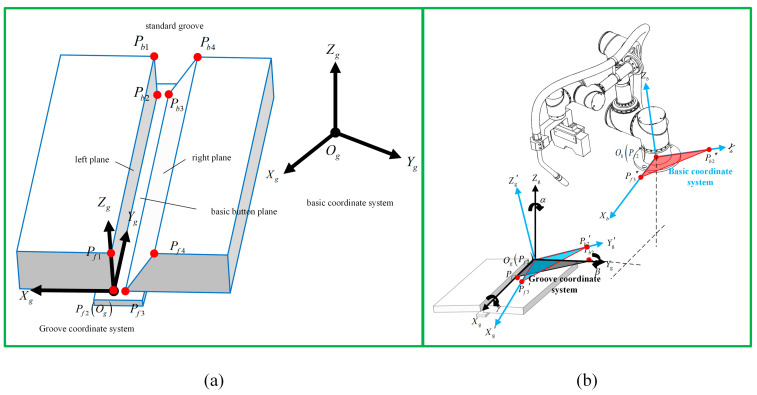
(**a**) The groove model and characteristic points are represented in the groove co-ordinate system. (**b**) Transformation relationship between the groove co-ordinate system and the basic co-ordinate system.

**Figure 6 materials-15-03796-f006:**
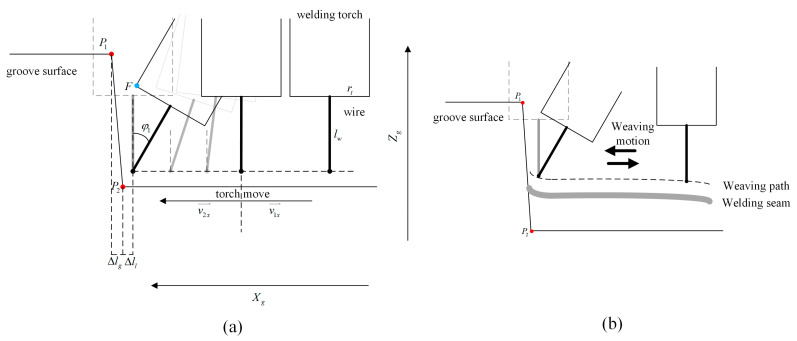
The weaving attitude of welding torch is adjusted to avoid obstacles. (**a**) The motion of the welding torch during welding of the first layer. (**b**) The motion of the welding torch during welding of the remaining layers.

**Figure 7 materials-15-03796-f007:**
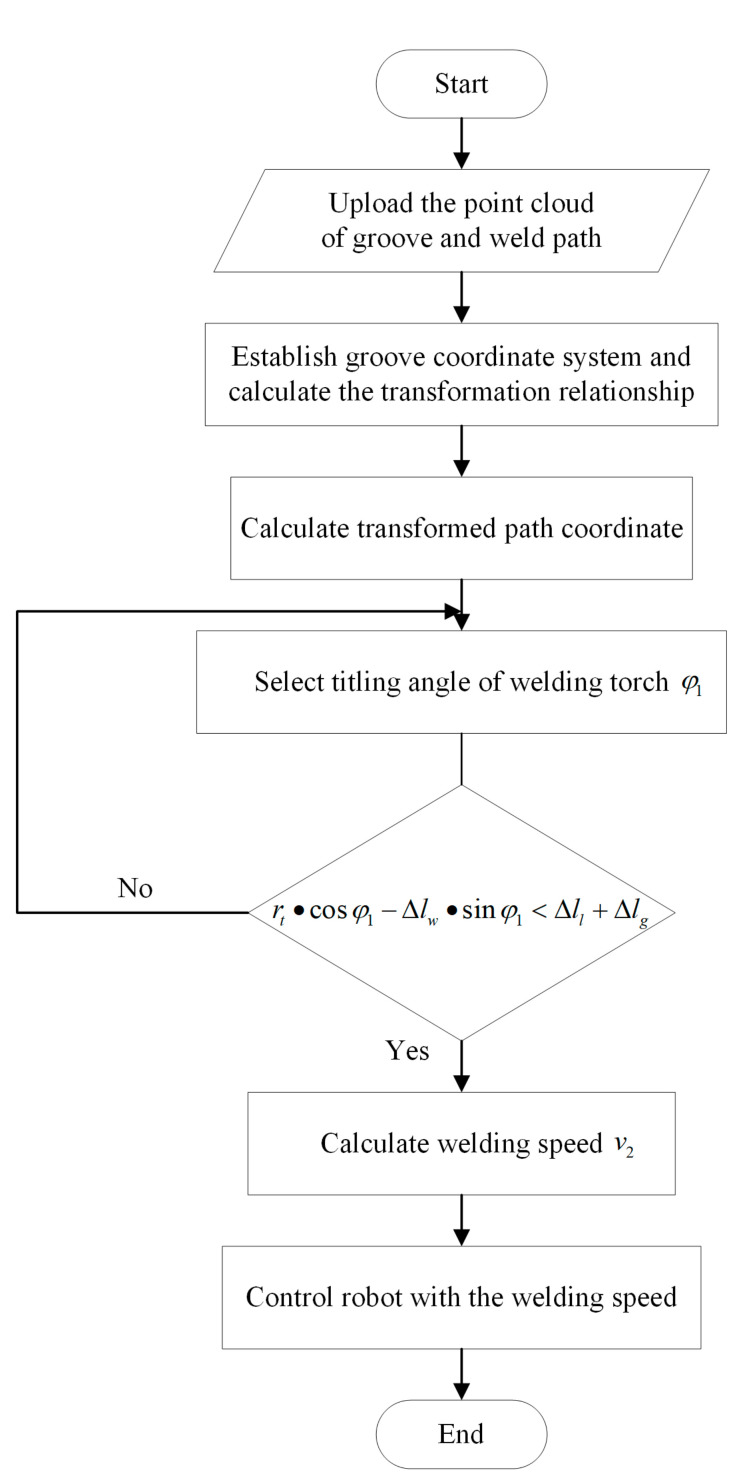
Welding planning flow chart using the proposed algorithm.

**Figure 8 materials-15-03796-f008:**
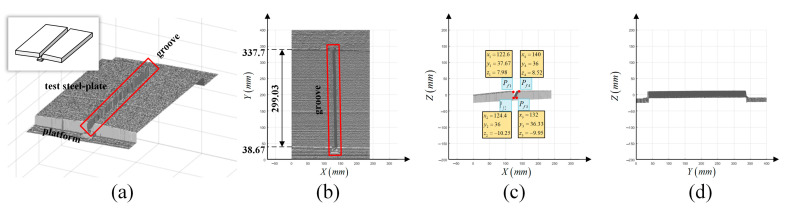
(**a**) Three-dimension reconstruction model of the test steel plate. The gray surface is the reconstructed three-dimension steel plate. The three-dimension model is displayed in the upper-left corner. (**b**–**d**) Projection of the model on plane *XOY*, *XOZ*, *YOZ*.

**Figure 9 materials-15-03796-f009:**
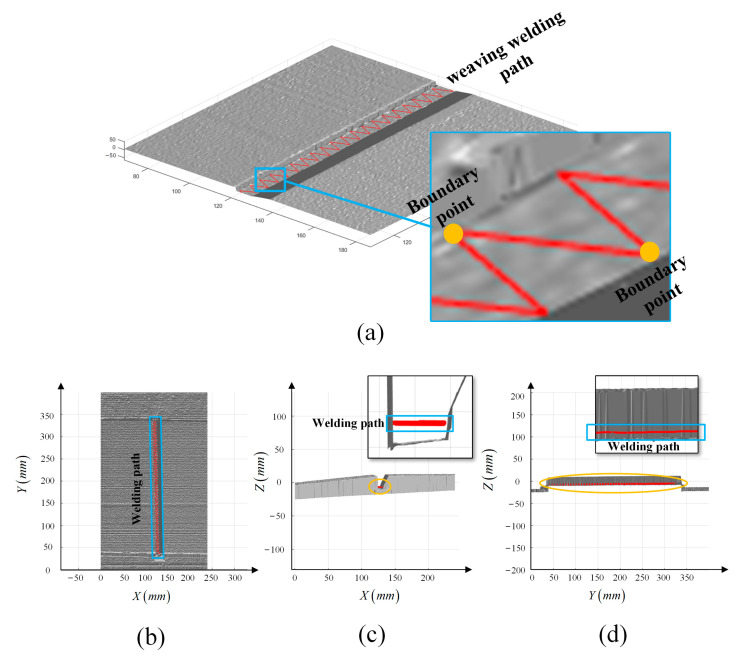
(**a**) Welding path in the groove; the red broken line is the planned welding path. The image in the blue box is a partial enlargement, and the yellow points in blue box are the boundary points of the welding path. (**b**–**d**) Projection of the model and planned weaving welding path on plane *XOY*, *XOZ*, *YOZ*.

**Figure 10 materials-15-03796-f010:**
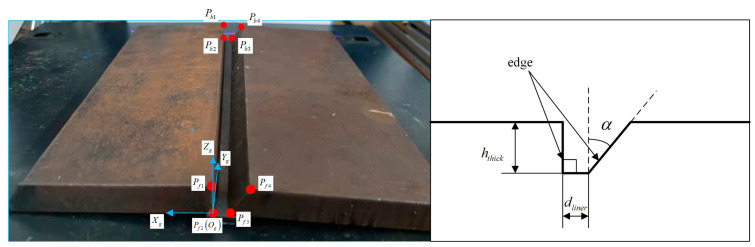
Characteristics points and parameters of the test steel plate. As for the selected groove, the thickness of the test steel-plate is represented by $h_{thick}$, and the width of the button face is represented by $d_{liner}$; the left edge is vertical, and the right edge forms an angle of α degrees. $h_{thick}$ is 20 mm, $d_{liner}$ is 6 mm and α is 40°. The blue arrows and red circles are groove co-ordinate system and characteristics points represented in Figure 5a.

**Figure 11 materials-15-03796-f011:**
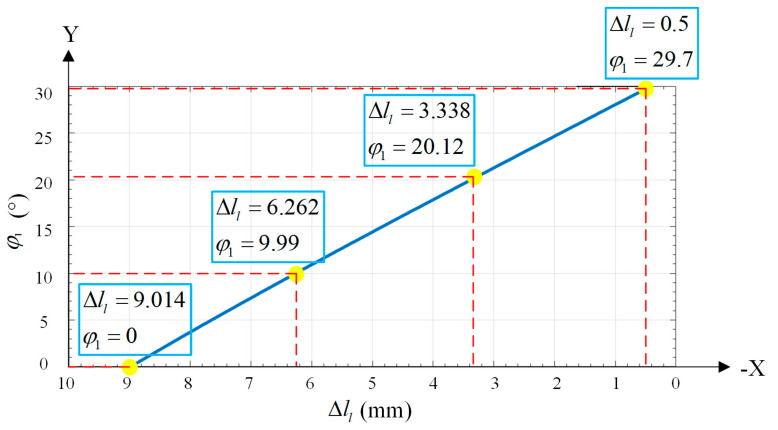
Welding torch inclination $\varphi_{1}$-$\Delta l_{l}$ curve after planning. Furthermore, the highlighted points represent the serval inclination states of the welding torch during the weaving process.

**Figure 12 materials-15-03796-f012:**
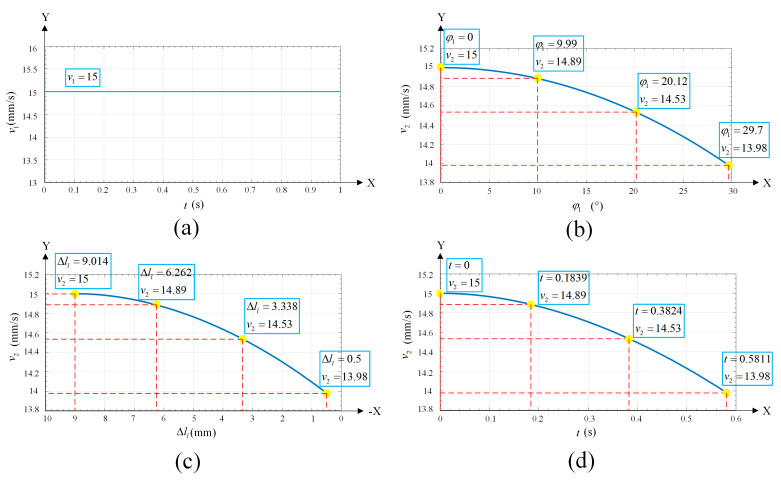
(**a**) The velocity curve before planning and the velocity is always 15 mm/s. (**b**) The velocity-$\varphi_{1}$ curve after planning. (**c**) The velocity-$\Delta l_{l}$ curve after planning. (**d**) The actual velocity curve after planning. The highlighted points in the figures represent the same states of inclination during the weaving welding process.

**Figure 13 materials-15-03796-f013:**
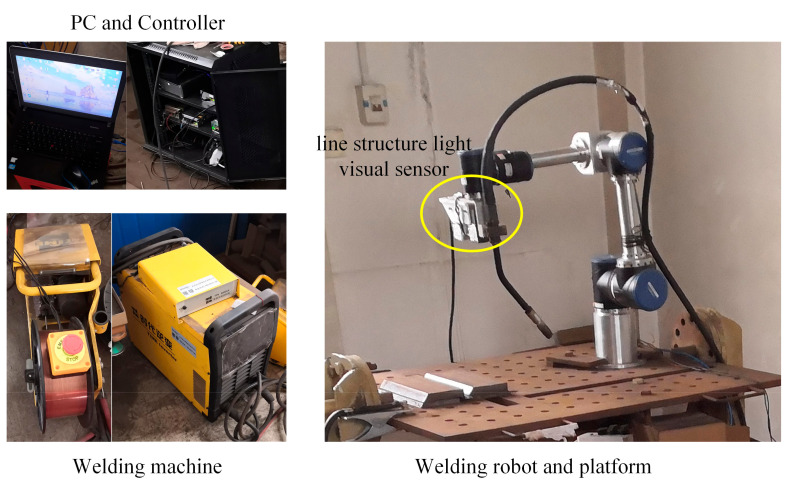
The robotic arc welding system.

**Figure 14 materials-15-03796-f014:**
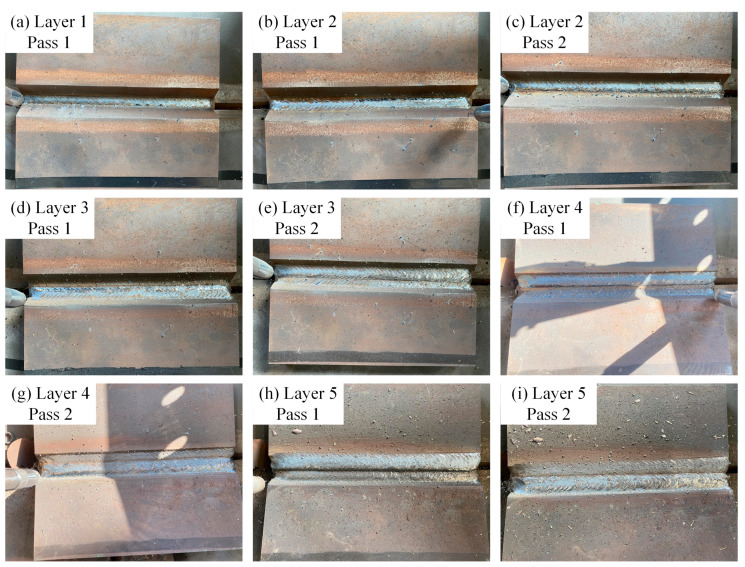
Actual welds of the experimental sample. (**a**) 1–1; (**b**) 2–1; (**c**) 2–2; (**d**) 3–1; (**e**) 3–2; (**f**) 4–1; (**g**) 4–2; (**h**) 5–1; (**i**) 5–2.

**Figure 15 materials-15-03796-f015:**
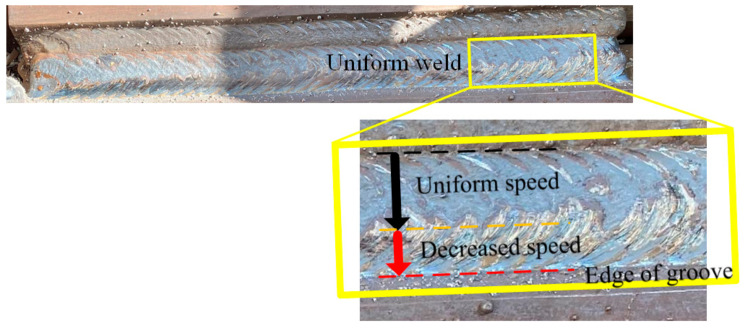
Selected weld and the partial features of the experimental sample.

**Table 1 materials-15-03796-t001:** Weaving welding parameters.

Layer-Pass Number i–j	Welding Current (A)	Welding Voltage (V)	Set Welding Line Speed (mm/s)	Dwell Time (ms)	Weaving Amplitude (mm)	Weaving Period Length (mm)	Welding Torch Inclination (°)
1–1	270	30	15	300	4	7	0~29.7
2–1	270	30	15	300	5	7	0~−29.7
2–2	270	30	15	300	6	7	0~29.7
3–1	270	30	15	300	7	7	0~−29.7
3–2	270	30	15	300	7	7	0~29.7
4–1	270	30	15	300	8	8	0~29.7
4–2	270	30	15	300	8	8	0~−29.7
5–1	270	30	15	300	9	9	0~29.7
5–2	270	30	15	300	9	9	0~−29.7

## Data Availability

Not applicable.

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
