# Peer review of "Towards a Uniform Welding Quality: A Novel Weaving Welding Control Algorithm Based on Constant Heat Input"

_materials, 2022, doi:10.3390/ma15113796_

Round 1
Reviewer 1 Report
The paper entitled ‘Towards a uniform welding quality: a novel weaving welding 2 control algorithm based on constant heat input’ deals with the development of weaving welding control algorithm targeting to retain the weld heat input constant by the velocity adaptive planning. The paper contains significant content for this scientific field. However, the reviewer thinks that several improvements should be carried out in order the paper to be published.
- The introduction should be enriched, and the scope of the paper should be state in a clearer way. The literature is poor and should be also enriched.
- The experimental part is well established but should be more extended and a correlation with the theorical approach should be cited.
- The conclusion part is poor and should be rewritten.
Reviewer 2 Report
The article is a good example of the use of heat transfer theory to control the thermal energy distribution in multi-layer welding with a robot.
Author Response
Thanks for your review
Reviewer 3 Report
Review report on the topic ‘Towards a uniform welding quality: a novel weaving welding control algorithm based on constant heat input’. Comments are listed below:
- Strengthen the abstract section. Add the key conclusion of the works in the last two lines of the abstract section.
- Discuss the motive behind the work. The clear application of the work should be discussed in the introduction section.
- There are numerous spelling and grammatical errors. Please revise the manuscript thoroughly. Sentences are also not complete and references are also cited in a rough manner.
- Try to make a bridge between current and previously published work and specify the gap area and objective of the work. Refer to following works: https://doi.org/10.1007/s13296-016-6007-z; https://doi.org/10.1007/s12540-020-00921-4.
- Introduction section is very length. Remove the unnecessary information.
- Mention the equation references.
- Also discuss the experimental section in detail.
- Work is good but there is no validation and this is serious flaw of the work.
- Simulation work needs more clear discussion along with boundary conditions and equations.
- In present, it looks like a technical report. The results are presented without any technical discussion.
Round 2
Reviewer 3 Report
Accepted